# Effects of the Interface Properties on the Performance of UV-C Photoresistors: Gallium Oxide as Case Study

**DOI:** 10.3390/s25020345

**Published:** 2025-01-09

**Authors:** Maura Pavesi, Antonella Parisini, Pietro Calvi, Alessio Bosio, Roberto Fornari

**Affiliations:** 1Department of Mathematical, Physical and Computer Sciences, University of Parma, Viale delle Scienze 7/A, 43124 Parma, Italy; antonella.parisini@unipr.it (A.P.); pietro.calvi@unipr.it (P.C.); alessio.bosio@unipr.it (A.B.); roberto.fornari1@unipr.it (R.F.); 2IMEM-CNR Institute, Viale delle Scienze 37/A, 43124 Parma, Italy

**Keywords:** SnO_2−x_/κ-Ga_2_O_3_ interface, contact resistance, TLM, four-point probe configuration, UV-C detectors, photogain, rise and fall times

## Abstract

Electrical contacts are of the greatest importance as they decisively contribute to the overall performance of photoresistors. Undoped κ-Ga_2_O_3_ is an ideal material for photoresistors with high performance in the UV-C spectral region thanks to its intrinsic solar blindness and extremely low dark current. The quality assessment of the contact/κ-Ga_2_O_3_ interface is therefore of paramount importance. The transfer length method is not applicable to undoped Ga_2_O_3_ because the interface with several metals shows a non-ohmic character, and a non-equivalent contact resistance could restrict its applicability. In this work, a new methodological procedure to evaluate the quality of contact interface and its effect on the sensing performance of UV-C photoresistors is presented, using the SnO_2−x_/κ-Ga_2_O_3_ contact interface as a case study. The proposed method includes a critical comparison between two-point and four-point probe measurements, over a wide range of voltages. The investigation showed that the effect of contact resistance is more pronounced at low voltages. The presented method can be easily extended to any kind of metal/semiconductor or degenerate-semiconductor/semiconductor interface.

## 1. Introduction

Photoconductive materials, selective in the UV range, received great attention owing to their noteworthy civil, scientific, and military applications [1]. The use of UV detectors in so many different fields requires the combination of high sensitivity, high selectivity, fast rise and fall times, compactness, robustness, flexibility, and compatibility with standard electronics. Examples of applications include the monitoring of the atmosphere, volcano plumes, ground irradiation, and human UV exposure (wearable sensors) [2,3,4,5]. Additionally, applications extend to the detection of free flames or electrical discharges caused by the damage of insulation in high-voltage lines [6,7], the monitoring of hydrogen combustion [8], and the implementation of systems for national safety [9], space communication [10], and UV astrophysics [11].

Recently, the ultrawide-bandgap semiconductor gallium oxide gained increasing attention as material very suitable for the detection of UV-C radiation (UV-C refers to the range of 100 ÷ 280 nm of UV radiation, very dangerous for skin cancer). The solar radiation in the range of 240 ÷ 280 nm is almost completely absorbed by the ozonosphere; hence, in normal conditions, such radiation is not detected at the ground level. The human eye does not percept the UV light, so the detection of this high-energy radiation from artificial sources or in ozone-depleted areas is crucial for human health. It is also important to detect a UV-C signal in daylight, and to this purpose, devices based on ultrawide-bandgap semiconductors offer some advantages over standard semiconductors as there is no need to filter the visible light. UV detectors are insensitive to low-energy background radiation and false alarms are rarely produced. This has stimulated an increased interest for military and civil applications of UV photodetectors [12].

In this scenario, the ultrawide-bandgap gallium oxide (Ga_2_O_3_) offers some favorable features: it is cost-effective, safe, and sustainable, despite its relatively long response time and slow recovery after light exposure caused by carrier trapping [13]. Among the different polymorphs of gallium oxide, the orthorhombic κ-Ga_2_O_3_ is particularly interesting because of its optical bandgap of about 4.8 eV [14] and a higher lattice symmetry with respect to the monoclinic β-phase, coupled to the high responsivity and solar blindness of devices based on this material. Our group started the research on the κ-polymorph some years ago and demonstrated that it is thermally stable up to 700 °C, it has higher crystallographic symmetry than the monoclinic β-Ga_2_O_3_, and it can be deposited on conventional substrates (like sapphire) at lower temperatures and milder epitaxial conditions. A complete conversion of the metastable κ-phase to the stable monoclinic β-phase occurs only above 900 °C. In addition, it exhibits symmetric electronic and vibrational properties, and no cleavage problems. These characteristics make the κ-phase of Ga_2_O_3_ cost-effective and interesting for non-critical operational conditions, such as for typical UV-C detection, and justifies the efforts made to overcome the residual shortcomings via the optimization of the growth parameters and the enhancement of the quality of interfaces [15,16,17,18].

A central issue for applying this material in large-scale device production is the formation of good and stable interfaces with metal contacts. This is a general challenge for ultrawide-bandgap materials, like gallium oxide, in which the formation of unwanted energy barriers at the metal–semiconductor interfaces can reduce the charge collection efficiency thus limiting the benefit of the high intrinsic photogain of the absorber material [16].

The detection of UV-C light by using gallium oxide was obtained both with the use of photodiodes and photoresistors. The time reaction of diodes is normally quicker than that of photoresistors. Furthermore, a photodiode can exploit the photovoltaic effect and operate in self-powered mode, but in the case of Ga_2_O_3_, where no p-doping is available, p-n junctions require a heterostructure that can adversely affect the carrier collection because of the point defects and traps associated with the interface, interdiffusion phenomena and possible interlayer formation. Schottky diodes also present issues and tend to fail at high temperatures [19,20,21]. The presence of trapping states at the surface or at the interfaces can drastically reduce the response speed. On the other hand, it should be noted that the intensity of incoming light, the contact geometry, and the applied voltage also play an important role on the photoresponse speed [22,23].

Photoresistors offer some advantages in terms of easier fabrication procedure and higher sensitivity thanks to the larger active volume available for carrier photogeneration. The improvement of contact interface quality is however crucial for the photoresistors, as good low-resistivity ohmic contacts are essential for the effective collection of photocarriers.

Attempts to improve the quality of the metal contact on the widely studied β-Ga_2_O_3_, especially on the intentionally n-type doped material, were already reported in the literature [24,25,26,27]. The performance of devices based on β-Ga_2_O_3_ is indeed limited by the quality of the semiconductor–metal interface. The most popular ohmic contact to β-Ga_2_O_3_ is provided by the double-layer Ti/Au. However, a high annealing temperature is required to enhance the Ti/Au–Ga_2_O_3_ contact via intermixing, which makes the process dependent on crystal orientation [28]. It was also reported that an interlayer between metal and semiconductor could improve the contact [29].

According to the Schottky–Mott theory, a good ohmic contact to an n-type semiconductor (such as the nominally undoped Ga_2_O_3_) is obtained if the work function of the metal matches the electron affinity of the semiconductor, but this is not a sufficient condition to guarantee the ohmicity of the contact since the status of the semiconductor surface can also be decisive [30,31]. The formation of a space-charge region under the metal, as a consequence of the presence of trap levels and/or interface states, is often responsible for the non-ohmic behavior in the current/voltage curves. Such space-charge distribution normally does not reach a steady-state regime in a short time [16]. In addition, the presence of a large density of bulk defects, far from the contact interfaces [32], can also have an effect on the limitation of the current flowing in the device even after long stabilization times.

Less is known about the metal–semiconductor interfaces in the case of the κ-Ga_2_O_3_ polymorph. The double-layer Ti/Au used for β-Ga_2_O_3_ was also adopted for κ-Ga_2_O_3_ without much critical insight [13,33,34]. The use of the Ti layer in Ga_2_O_3_ enhances the adhesion of gold onto the semiconductor surface and reduces the Schottky barrier height only after a post-deposition annealing. Unfortunately, high-temperature thermal treatments are not applicable to the metastable κ-phase [35]. An acceptable and reproducible ohmic behavior with no thermal treatments so far was obtained for doped materials [13] but not for the high-resistivity material needed for photoresistor fabrication.

Different metallization schemes, such as ZnO/Ti/Au or ITO/Pt, fabricated on nominally undoped κ-Ga_2_O_3_ films, did not exhibit linear current–voltage (I–V) characteristics at any temperature [36]; heating at moderate temperatures under inert gas for a long enough time may help to improve the characteristics of contacts by lowering the barrier height at the interface. Alternatively, inserting a heavily doped region between the metal and semiconductor can reduce the barrier at the interface and provide “in situ” ohmic contacts without annealing [37].

Recently, a novel procedure has been proposed to obtain ohmic contacts on κ-Ga_2_O_3_ by depositing a SnO_2−x_-ITO bilayer [36]. The procedure does not involve high-temperature steps, as required to preserve the properties of this metastable polymorph [38].

In the present study, an in-depth investigation of the SnO_2−x_/κ-Ga_2_O_3_ interface has been undertaken with the aim of minimizing the bias applied to a planar photoresistor; this reduction could be eventually achieved by employing contacts shaped like an inter-digitated pattern. We present here a novel approach suitable to evaluate the contact resistance when the transfer length method (TLM) is not applicable [39], for example, in the case of non-ohmic behavior or when the slope of the I–V curves does not scale with the distance between contacts. As a case study, this methodology has been developed for undoped κ-Ga_2_O_3,_ but this method could readily be applied to other wide-bandgap semiconductors. The device selected as a case study was a photoresistor based on κ-Ga_2_O_3_ with characteristics sufficiently good to verify the applicability of the new approach as well as its ability to determine the voltage range and contact geometry that give the highest performance. UV-C photoresistors based on this active material demonstrated a rejection ratio of UV/VIS higher than 104 and a high spectral responsivity as was already reported in [16]. The presence of a contact resistance, even under illumination, has been evidenced at the SnO_2−x_/κ-Ga_2_O_3_ interface; its value changes with bias, so that at low voltages it overcomes the material resistance. Contact resistance is detrimental to the performance of UV-C photoresistors, as it reduces the photogain (i.e., the responsivity) and increases the response time, particularly when the distance between contacts decreases.

## 2. Materials and Methods

The thin films of Ga_2_O_3_ used for this study were grown by low-pressure MOCVD at 60 mbar on 2-inch c-plane sapphire substrates (supplied by Cryscore, Jiaozuo, Henan, China) heated at 650 °C, with He as a carrier gas. Trimethylgallium (TMG) and ultrapure water were used as precursors (supplied by Dockweiler Chemicals GmbH—Marburg, Germany), typically with a H_2_O/TMG ratio range of 100 ÷ 350. Nominally undoped pure κ-phase Ga_2_O_3_ films, transparent and smooth, with a thickness of about 500 nm were selected. In order to standardize the surface condition, the films were first etched in a solution of hydrofluoric and nitric acid (HF 50% + HNO_3_ 50%) for 1 min and then sequentially rinsed in acetone, isopropanol, deionized water, and finally dried in dry nitrogen (chemicals were supplied by Carlo Erba Reagents, Milano, Italy).

A one-face planar geometry, reproducing a TLM pattern, was chosen for electrical measurements in order to evaluate the contact resistance by means of the TLM. Five contacts, numbered from 1 to 5, with an area of (0.4 × 0.04) cm^2^ were deposited through a stencil metallic mask, with increasing distances L between adjacent contacts (L_1_ = 200 μm, L_2_ = 400 μm, L_3_ = 800 μm, and L_4_ = 1600 μm), with contacts 1 and 2 being the closest (Figure 1a). A tolerance of about 5% on the distances between contacts was confirmed by optical microscope.

The SnO_2−x_ metal contacts were deposited by a 13.56 MHz radio-frequency magnetron sputtering in an Ar + O_2_ environment at room temperature, starting from a 3-inch Sn target with 6N purity. Direct Sn deposition cannot be used because it leads to the formation of droplets on the surface at deposition temperatures higher than the melting point of Sn (220 °C). By depositing a tin-rich oxide film on the surface of the semiconductor, it is possible, instead, to produce an effective n-type doped layer under the contact pad via the diffusion of Sn atoms.

The films are amorphous, and the oxygen content can be tuned (0 ≤ x ≤ 1) by varying the sputtering power density and the partial pressure of O_2_. The parameters of deposition were optimized and a power density of 0.7 Wcm^−2^, under Argon backpressure of 0.3 Pa, was chosen to obtain both a good adherence and a good electrical response (the thickness of SnO_2−x_ is a few tens of nm) [38]. Slightly Sn-rich films were obtained, characterized by an energy gap of 3.56 eV, a carrier concentration of 1.25 × 10^20^ cm^−3^, and an electron mobility in the range of 1 ÷ 5 cm²/Vs. Additionally, the mobility increases up to 15 cm²/Vs with increasing oxygen content in the sputtering chamber. These properties correspond to a resistivity in the range of (5 ÷ 1) × 10^−2^ Ωcm.

Films showed a good optical absorption in the UV-C region with a marked suppression above 270 nm [32], whereas the percentage of UV light transmitted under the contacts is negligible.

A previous High-Resolution Transmission Electron Microscopy (HRTEM) study [40] showed that no significant change in the film surface was introduced by SnO_2−x_ sputtering. The HRTEM images reported, along with the results of ToF-SIMS and RBS investigations, showed however that sputtering deposition could promote the formation of a very thin intermixing layer, due to the interaction between Sn atoms and the surface of the gallium oxide epilayer.

The current–voltage (I–V) characteristics of each pair of adjacent electrodes were recorded with a Source-Meter Keithley Mod. 2400 up to 200 V, both in dark and under illumination with different photon fluxes, with energy tuned at the bandgap of the material (254 nm). Photon flux from a Wood lamp (Spectroline E-Series Ultraviolet Hand Lamp, Model ENF-280C, supplied by Merck, Darmstadt, Germany) was varied by using neutral filters.

A preliminary use of the TLM approach did not yield significant results; consequently, current–voltage measurements under light at the bandgap energy have been performed in two different configurations, i.e., two-point and four-point probe measurements.

The two-point probe configuration is the most common way to record I–V curves (Figure 1b); this method uses the same pair of contacts both for injecting current and for measuring voltage. In cases where the resistance of contacts becomes comparable to the sample resistance, a four-point probe method is to be used in order to distinguish the resistance contribution of contacts from that of the photoactive material.

The four-point probe method circumvents the contribution of the contact resistances by injecting current through the outer pair of contacts and measuring the voltage between the inner pair (Figure 1c). In this way, a constant current streams along the sample, between the contacts labeled 2 and 5, resulting in a voltage drop between the two inner probes (contacts labeled 3 and 4). As there is no current flowing through the metal/semiconductor interfaces of the inner pair of contacts, the measured voltage only derives from the material resistance.

The four-point probe measurements were conducted with a Keithley 220 Programmable Current Source, which features an output resistance exceeding 10^14^ Ω (on 1nA range) and a Keithley 617 Electrometer. The input impedances of instruments involved in voltage measurement are much higher than measured resistances (Source Meter Keithley 2400: >10 GΩ, Electrometer Keithley 617: >200 TΩ).

Two-point and four-point probe measurements allowed to calculate the contact resistance under illumination as a function of the applied voltage, on the basis of the electric model of Figure 1b,c. The effects of the presence of the contact resistance on the performance of the UV-C photoresistors were then highlighted, by making explicit the photogain value in the different injected current ranges and performing photocurrent measures with cycles of turning the light on and off for different distances between contact pads. A mechanical shutter was used to modulate the light directed to the samples and obtain on–off illumination. The off period was shorter than the characteristic photoresponse time of the investigated devices. The time transients after turning the light on and off were acquired with a 66 Hz sampling frequency.

## 3. Results and Discussion

### 3.1. Electrical Characterization by TLM in Dark and Under Illumination

A preliminary application of the TLM was made on κ-Ga_2_O_3_ epilayers with five SnO_2−x_ contacts (labeled from 1 to 5), both in dark and under illumination. It is well known that metal oxide films exhibit relatively slow response times to electrical/optical excitations due to processes such as charge carrier trapping/detrapping and/or not perfect ohmicity of semiconductor–metal interfaces. Taking this into account, the system was allowed to relax for three minutes after each voltage-increase step. This means that the electrical I–V value pairs were recorded after the completion of the transient following each V increase step, i.e., in quasi-stationary condition.

In Figure 2a, we report the I–V curves in dark for a representative sample: an ohmic trend can be recognized, but there is no correlation between current values and contact spacing. Calculated resistances are on the order of 10^10^ Ω. In agreement with Ohm’s law, the resistance should vary with the distance between electrodes if the film is uniform in thickness and background doping. However, here this trend is not respected, in particular for the pair 1–2. In addition, a weak hysteresis in the I–V curves was observed when executing a cycle over the full voltage range. This result suggests that the contact resistance is comparable with the one of the semiconductor, whose value only depends on the distance between electrodes, but it is different for the five couples of electrodes in contrast with the requirement of the TLM.

The same electrical measurements under illumination at 254 nm provide an opportunity to explore the applicability of the TLM approach. In Figure 2b, the I–V curves under illumination show a linear trend, but the TLM analysis (Figure 2c) does not give reliable results because the resistance between contacts, as a function of their distance, deviates from linearity. This result, confirmed on several samples, could be due to a reduced charge collection efficiency, particularly corresponding to larger contact spacing.

A TLM analysis [39], restricted to the closest pairs of contacts (Figure 2c, red dashed line), provides a sheet resistance of about 1.84 × 10^9^ Ω, but the accuracy is not sufficient to obtain the value of the contact resistance. A resistivity of about 9.2 × 10^4^ Ω cm has been estimated, by considering the layer thickness of 500 nm.

In conclusion, when dealing with high-resistance material, the contact resistance and the strong contribution from trapping make the TLM inappropriate and, for this reason, it is mandatory to eliminate the influence of the contact resistance during the I–V measures by means of a four-point configuration.

### 3.2. Two-Point and Four-Point Probe Electrical Characterization Under Illumination

To compare the results obtained with the two-point and four-point probe configurations, it is more convenient to control the injected current (variable) rather than the applied voltage (function of the variable). Preliminary measurements were conducted to verify that the control of the current or the voltage leads to equivalent current–voltage curves.

Measurements in the two-point probe configuration have been taken on the pair of contacts with a distance of 800 μm (electrodes 3 and 4) in a wide range of injected currents under illumination at 254 nm with different irradiances. All the investigated structures exhibited characteristics with good linearity and symmetry, with respect to negative and positive currents, to the value of current, and to the photon flux (Figure 3).

In Figure 3b, the photocurrent value at 200 V shows a good linearity as a function of the incident photon flux, suggesting that the interfaces below the contacts do not affect the current flowing at high voltages under illumination.

To investigate the trend at low voltages, photovoltage values for the two-point configuration were acquired under light, by varying the injected current from zero to positive values, then returning back to zero, and repeating the sequence toward negative values and back (Figure 4a).

An evident deviation from the linear behavior is observed in the voltage range of 0 ÷ 2V, revealing a strong limitation in the current flow. As the measure has been conducted by controlling the injected current, such a trend at low voltage must be interpreted in terms of a voltage drop at the metal–semiconductor interfaces; in addition, the loop-like sequence shows that a more or less pronounced hysteresis is present for each value of the photon flux.

The same range of current was investigated in the four-point probe configuration for the same pair of contacts, labeled 3 and 4 (Figure 4b). In this case, the I–V characteristics exhibit good linearity and no hysteresis effects in the loop were observed at any photon flux level. This result was attributed to the decoupling of voltage–current circuits, as described in the Section 2.

The comparison of results suggests the presence of a remarkable effect of contacts in the two-point probe configuration, consisting of a limitation in the current injection at low voltages and relatively strong dependence on the intensity and duration of illumination. The presence of a space-charge region at the SnO_2−x_/κ-Ga_2_O_3_ interfaces due to traps, changing their state of charge with the time, and as a function of illumination, could be a plausible interpretation of this experimental evidence. In the four-point configuration, there is no current flowing through the interfaces between semiconductor and electrodes 3 and 4; therefore, the contribution of contacts is negligible. This is also consistent with the absence of hysteresis in the current–voltage loops.

Referring to the equivalent electrical circuits of Figure 1c, in the four-point probe configurations, only the film resistance R is present between contacts 3 and 4, whereas in the two-point probe configuration, the film resistance R is in series to the contact resistances, each assumed to be equal to R_C_ (Figure 1b). For each value of injected current, it is then possible to calculate R and R_C_.

The results of calculation, made on the basis of these equivalent circuits (Figure 5a), are shown in Figure 5b for data acquired at the maximum optical power (50 µW/cm^2^).

The prevalence of the contact resistance over the film resistance is found for injected current values below 5 nA (i.e., below 2V), as expected. The drop of the contact resistance around zero, approaching the lowest values of current (below 1 nA), could derive from gallium oxide surface conduction or from the residual electrical polarization of the material. The film photoresistance R is almost independent of the injected current and equal to 1.53 × 10^8^ Ω; assuming a current penetration depth of about 500 nm, a resistivity of about 3.82 × 10^4^ Ωcm at an optical power of ~50 µW/cm^2^ can be estimated. This value differs quite significantly from the value obtained by the TLM (Figure 2c); we believe that in the TLM approach, the discrepancy can derive, for example, from a contact resistance that differs from electrode to electrode and/or from edge effects.

The contact resistance Rc, instead, depends on the injected current for values below 5 nA (see Figure 5b), reaching few 10^8^ Ω when the current is close to zero. This result again suggests the presence of a potential barrier between the SnO_2−x_ contact and the film. The same curves have been analyzed in a wide range of injected currents and reported on a logarithmic scale (Figure 6), for both two-point probe and four-point probe configurations. Three conduction regimes have been evidenced.

For an injected current below 4 nA (low-current region), the photovoltage shows very little variation for both configurations; in the mid-current region (from 4 nA up to 1 µA), the four-point probe curve is nearly ohmic, while the two-point probe curve shows a drastic change in slope with a flex at about 10 nA as a consequence of the abrupt decrease of R_C_ for increasing current, already shown in Figure 5b. Finally, in the high-current region above 1 µA, the curves run parallelly for both measurement configurations, with a slope that approaches unity as expected for an ohmic regime of conduction.

The comparison of curves in Figure 6 confirms the critical role of the interface between SnO_2−x_ and semiconductor in the two-point probe configuration. A tentative model that can justify the three conduction regimes for the two-point probe configuration includes the following elements: in the low-current region (<4 nA), the conduction occurs at the surface, supported by a thin layer of electron accumulation, as also predicted for the β-phase [41]. As an alternative interpretation, the current could flow within a very thin surface layer of semiconductor characterized by efficient trap-assisted transport, until all the available traps are involved in the transport of charge carriers. Toward the end of the low-current region, the photovoltage starts to increase but the current is nearly constant since the thin surface channel of conduction is saturated and the conduction within the film of Ga_2_O_3_ is inhibited by the contact barriers. In the mid-current regime, for injected current higher than 8 nA, the influence of the surface conduction and contact barrier diminishes and the I–V characteristic in a Log–Log scale exhibits a slope change, approaching a linear behavior with a slope of about 2.49. The contact resistance indeed limits the conduction until a voltage drop of about 1V is reached; beyond this voltage, a space-charge-limited conduction is likely to occur, which justifies the slope much larger than one for an exponential distribution of traps [42]. At injected currents higher than about 1 µA, the curve approaches the ohmic behavior and becomes very similar in slope to the curve acquired in the four-point probe configuration.

The contact resistance shows its major influence at low voltages (Figure 5a), but the reduction in current flowing in the two-point probe configuration, with respect to the four-point probe configuration, is observed also at high voltages (note that the two characteristics have the same slope but a two-point current that is slightly lower than the four-point current must be supplied to achieve the same voltage in the entire high-current range; see Figure 6). The presence of a contact resistance dramatically impacts the performance of the UV-C resistive sensors, particularly the photogain and the photoresponse with time.

### 3.3. Effects of Interface SnO_2−x_/κ-Ga_2_O_3_: Photogain and Rise Time in the UV-C Photoresistors

The photogain *G* is an important parameter for photodetectors, indicating the ratio of collected electrons *N_el_* with respect to absorbed photons *N_ph,abs_*, and it is directly related to the photocurrent *I_PC_*. As reported in [16]:(1)$$G=\frac{N_{el}}{N_{ph,abs}}=\frac{I_{PC}}{A\cdot P_{opt}}\cdot \frac{hc}{e\lambda }$$
where *A* is the active area of the photoresistor exposed to the light, and *P_opt_* is the optical power, assumed to be completely absorbed by the photoactive material. The quantities *h*, *c*, *e*, and *λ* are the Planck constant, the speed of light, the electron charge, and the photon wavelength, respectively. The photogain for illumination at the bandgap wavelength, calculated from Equation (1) starting from the data of Figure 6, is plotted in Figure 7a for the pair of contacts 3–4 (black curves, contact distance of 800 μm), for both the two-point and the four-point measurements.

For comparison, the curves acquired for the pair of contacts 2–3 are also reported (red curves, contact distance of 400 μm). The contact resistance affects the two-point probe measurements for both pairs in the same way; the photogain is reduced up to one order of magnitude in the low-voltage region (<5V) with respect to the four-point probe measurement. Furthermore, at higher voltages, the difference between the two configurations is more appreciable by reducing the distances between contacts. The slopes of curves for the same configuration are very similar independent of contact spacing. The green dashed line marks the unitary value for the photogain [16], which is exceeded only for voltages higher than 8V and 20V for the pairs of contacts 2–3 and 3–4, respectively. Below such values, the trapping and the contact barrier effects dominate the conduction.

Photoconduction theory assumes a linear ohmic dependence of the photogain on the ratio between voltage and contact spacing for adjacent contacts [43], considering a uniform electric field along the conduction channel and a conduction from extended states. Figure 7b shows the photogain of Figure 7a as a function of V/L^2^: the linear trend predicted by the theory is observed only corresponding to voltages higher than 40V and 35V for the pairs of contacts 2–3 and 3–4, respectively. On the other hand, below these specific voltage thresholds, the photogain remains significantly below one. This limitation occurs because of the low lifetime of carriers (recombination mechanisms) but also due to the interface effects described with regard to Figure 6. The observed dependence of photogain on the contact spacing also suggests that reducing the distance between the electrodes is not always advantageous in terms of detector efficiency. The contact resistance calculated for the pair 2–3 is comparable with the one calculated for the pair 3–4; however, its influence could be higher because the material resistance decreases. This fact must be considered when looking for the most convenient efficiency or applied voltage in photoresistors.

To complete the investigation on the contact–semiconductor interfaces and their effect on the performance of photoresistors, the time-dependent photoresponse for different contact spacing was investigated by performing on–off illumination cycles with UV light (254 nm). The normalized photocurrent for three subsequent on–off cycles is plotted in Figure 8a for the four contact distances of the TLM pattern in a two-point probe configuration by fixing the ratio between applied voltage and contact distance at 0.125 V/μm for each pair of adjacent contacts.

The stabilization of the response to the illumination requires at least three cycles; after that, a clear dependence of time response on contact spacing is evidenced. The same dependence is not observed when the light is switched off.

Usually, the evolution of photocurrent in time is analyzed by an exponential fit, but it is not easy to attribute physical meaning to the fitting time constants. More significant quantities, from the point of view of device performance, are the rise time and the fall time. The rise times t_ON_ (or the fall times t_OFF_) for the photoresistor are taken as the time lapse necessary to go from 10% to 90% of the saturation photocurrent in stationary condition (or the time lapse to drop from 90% to 10% of the saturation photocurrent).

Rise times and fall times for all pairs of contacts for the third cycle are reported in the insert of Figure 8c. The rise time decreases with an almost exponential trend with increasing contact distance: it decreases by about 50% corresponding to a four-fold distance increase. The interface between contact and semiconductor heavily affects the rise time so that again there is a trade-off between contact spacing and detector performance. A reduction in distance between contacts may help collection but may worsen the time response. The fall time, instead, does not show a clear dependence on the distance between contacts, and its behavior is more likely controlled by traps in the photoactive material that might be localized at domain boundaries or at the surface of films. A latency of photocurrent after turning the illumination off is evidenced only for the pair of contacts 3–4 (green curve in Figure 8a,b, probably due to the inhomogeneity of surfaces under contacts).

The transient rise time as a function of contact spacing follows a monotonic trend (fitted with an exponential curve in Figure 8c, green dashed line) that cannot simply be explained by considering the ohmic equivalent circuit used in the steady-state regime. Immediately after illumination, the resistance of the active layer of Ga_2_O_3_ decreases significantly due to carrier photogeneration, which, for a given applied voltage, results in a modified voltage drop at photoactive material and contact regions. The photocurrent transient under illumination depends on the settlement of the space charge under the contacts, up to reaching the final band bending. This evolution must be treated considering that contacts behave as an impedance rather than pure resistance. Therefore, the shorter the contact distance, the stronger its effect on the transient because the specific contribution of the two contacts to the total circuit impedance is higher.

The role of the contacts on photoresponse speed is also demonstrated by the different results obtained using the same Ga_2_O_3_ absorber but with Ti/Au contacts. Despite the good linearity over a wide bias range (Appendix A in Appendix A), the response time to UV illumination for the same pair of contacts is longer than for SnO_2−x_ (Appendix A). This suggests that for some applications, the SnO_2−x_-ITO contact system may be preferable to the Ti/Au one.

## 4. Conclusions

A careful investigation of the SnO_2−x_/κ-Ga_2_O_3_ interface has been undertaken with two primary objectives: (1) to explore innovative recipes for ohmic contacts to κ-Ga_2_O_3_; (2) to develop a general methodology suitable for the characterization of ohmic contacts in highly resistive wide-bandgap semiconductors, as an alternative to the TLM when it results are inapplicable.

Within this frame, photoresistors based on the epitaxial layer of undoped κ-Ga_2_O_3_ with SnO_2−x_ contacts have been investigated as a case study, and the results of I–V characterization using two-point or four-point measurement configuration were compared. Electrical measurements, at low-current regimes, both in dark and under illumination with photon energy above bandgap, reveal the presence of a contact resistance larger than the material’s own resistance. This phenomenon is likely due to the existence of a potential barrier at the interface.

This results in a strong limitation of the photocurrent collected at the electrodes in the voltage range below 2V while the photogain determined with the two-point probe measurement is consistently lower than the one from four-point measurements up to 10V (that corresponds to an injected current of about 1 μA). Beyond this threshold, in the so-called high-current regime, the two measurement configurations provide the same results. Photoresponse rise times are influenced by the contact resistance, with a major impact observed when the distance between the electrodes decreases. In contrast, fall times are primarily determined by the quality of active material or by its surface characteristics. This methodological study paves the way to the investigation and characterization of other contact technologies, with specific reference to the contact–semiconductor interfaces. Particularly when the conventional TLM does not provide accurate results, it is convenient to apply the standard I–V characterization, but with some precaution in the result interpretation. The comparison of 2-point and 4-point measurements may shed some light on the role of the contact–semiconductor interface and relevant trapping and space-charge mechanisms. For gallium oxide photoresistors, the optimization of the interface, through which the collection of carriers takes place, is mandatory. Only in this way, the reduction in the contact distances obtained via photolithographic interdigitated patterning may turn out to be beneficial, along with the appropriate bias range, to maximize the sensitivity and responsivity of photodetectors.

## Figures and Tables

**Figure 1 sensors-25-00345-f001:**
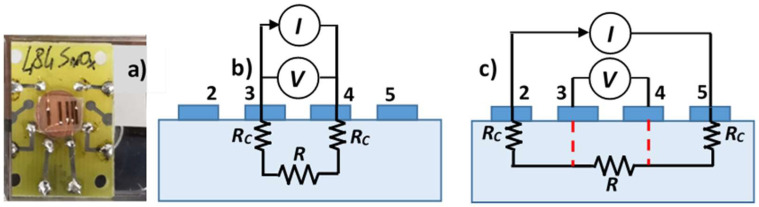
(**a**) Device with a TLM pattern; (**b**) two-point probe and (**c**) four-point probe configurations used for acquisition of I–V curves.

**Figure 2 sensors-25-00345-f002:**
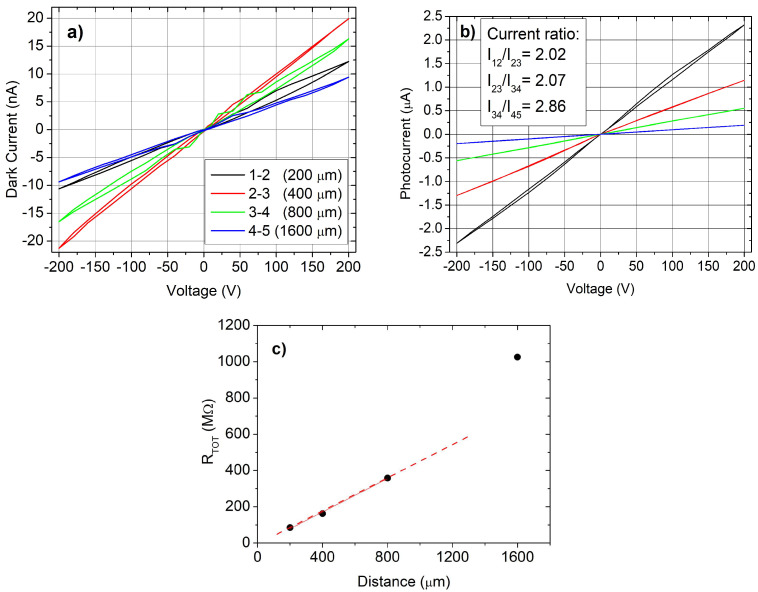
I–V curves for pairs of adjacent contacts: (**a**) without illumination (dark current); (**b**) under illumination at 254 nm (50 μW/cm^2^); (**c**) the total resistance under illumination as a function of the distance between electrodes; the TLM analysis [39] has been applied only to the initial linear trend (red dashed line).

**Figure 3 sensors-25-00345-f003:**
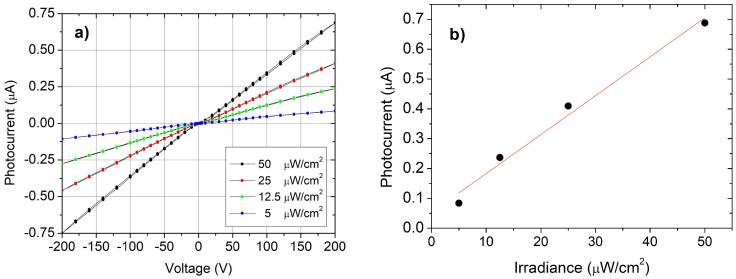
(**a**) I–V curves for electrodes 3 and 4 (800 µm) in the two-point probe configuration at different photon fluxes at 254 nm; (**b**) the photocurrent value at 200 V as a function of the incident irradiance.

**Figure 4 sensors-25-00345-f004:**
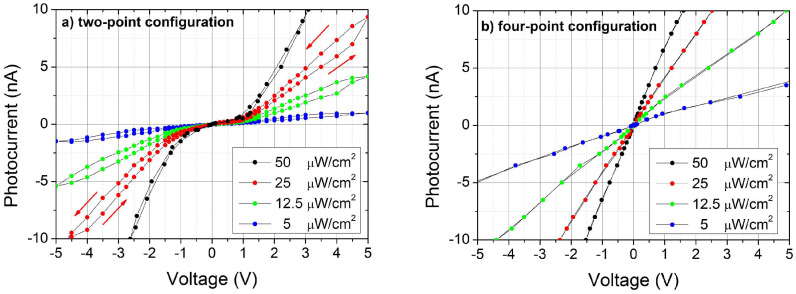
(**a**) I–V curves in the two-point probe configuration at different photon fluxes (photon wavelength is 254 nm). Values have been acquired following a loop sequence as indicated by red arrows; (**b**) the curves for the same pair of contacts (labeled 3 and 4) acquired in the four-point probe configuration.

**Figure 5 sensors-25-00345-f005:**
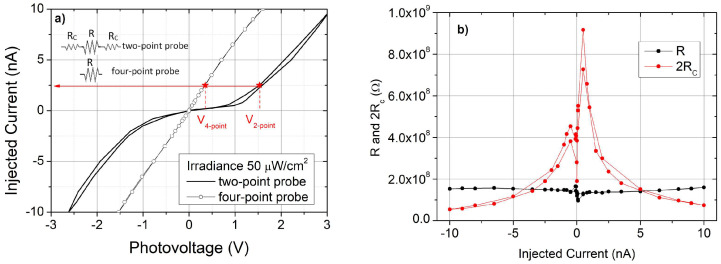
(**a**) I–V curves in the two-point probe and four-point probe configurations under the maximum irradiance at 254 nm (50 µW/cm^2^); (**b**) values of material resistance R and contact resistance 2 R_C_ as a function of injected current, calculated from data reported in (**a**) under the hypothesis of the equivalent electrical circuits therein depicted.

**Figure 6 sensors-25-00345-f006:**
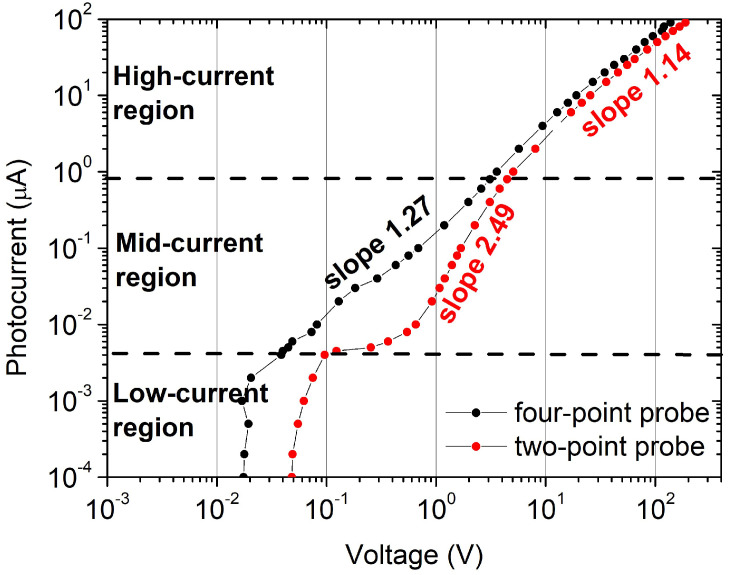
I–V curves in the two-point probe and four-point probe configurations under the maximum irradiance at 254 nm. Three different regions can be evidenced: (1) low-current region from 100 pA to 4 nA; (2) mid-current region from 4 nA to 1 µA; (3) high-current region for currents higher than 1 µA. The slopes are the exponents of the power law of current vs. voltage.

**Figure 7 sensors-25-00345-f007:**
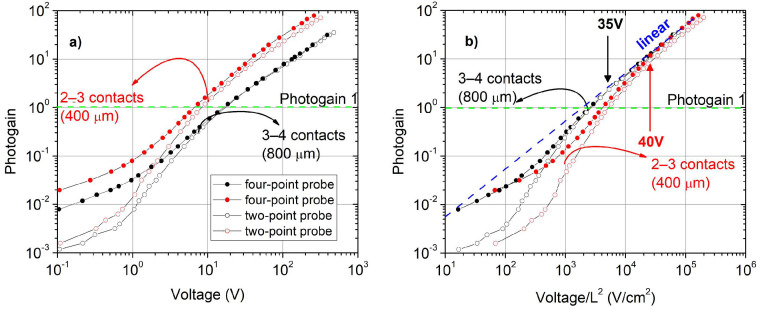
(**a**) Comparison of photogain for contact distances of 800 μm (pair of contacts 3–4, black symbols) and 400 μm (pair of contacts 2–3, red symbols) in both configurations, i.e., two-point probe (open circles) and four-point probe (full circles); (**b**) the same curves of (**a**) plotted as a function of V/L^2^. The blue dashed line represents the linear behavior and the green dashed line evidences the unitary value for the photogain. Lower voltage limits for the linear ohmic conduction region are indicated by red and black arrows in Figure 7b.

**Figure 8 sensors-25-00345-f008:**
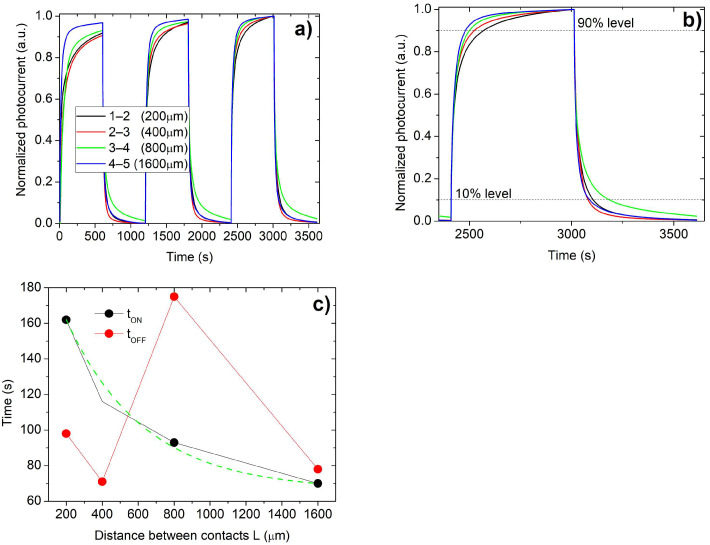
(**a**) Evaluation of time of photoresponse for the TLM pattern by repeating three cycles of on–off illumination sequences with photons at 254 nm. The irradiance is set at 50 μW/cm^2^ and the ratio between applied voltage and contact distance for each pair of adjacent contacts is the same (0.125 V/μm). (**b**) Time-dependent photoresponse for the third cycle of Figure 8a and the determination of rise time t_ON_ and fall time t_OFF_ from 10% and 90% levels. (**c**) Rise time and fall time as a function of the distance between contacts (green dashed line is the result of an exponential fit).

## Data Availability

Data are contained in the article.

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
