# Peer review of "Effects of the Interface Properties on the Performance of UV-C Photoresistors: Gallium Oxide as Case Study"

_sensors, 2025, doi:10.3390/s25020345_

Round 1

Reviewer 1 Report

Comments and Suggestions for Authors

Reviewer’s comments to authors: The manuscript fabricates a Ga2O3-based photoresistor for achieving UV-C by utilizing SnOx at the interface. The authors conducted only electrical measurements. They should also measure the structural and chemical properties, which are very important to design a high-performance photodetector. Overall, the article is not well-organized, and its presentation is also not good. The figure quality and resolution are very poor. Detailed comments are given below.

1.     The introduction of important concepts in the article must be very rigorous. For the introduction of UV photodetector based on PN junction and MSM, some more references should be added, for instance: (1) Materials & Design 2022, 221, 110917, (2) Advanced Electronic Materials, 2022, 8, 2200392. (3) Journal of Materials Research and Technology 2023, 22, 2174, (4) Solar Energy 2018, 174, 231–239.

2.     Authors have chosen k-Ga2O3 but it is found that it is not that much stable. The authors need to add more explanation as to why k-Ga2O3 is more important and useful instead of β-Ga2O3.

3.     Interface is very important for a photodetector because of charge transport and interfacial defect deteriorates the charge transport. There is a recent research work on the improvement of interface quality of Ga2O3-based diode: Appl. Phys. Lett. 2024, 124, 262101. Authors should compare their diode characteristics with those reported in a Table.

4.     Authors should add a digital image of the device having a TLM pattern on it.

5.     The rise and fall times of the SnOx/Ga2O3 photoresistor are in the range of seconds, whereas the current photodetectors based on Ga2O3 show ms rise time. The authors add some discussion on it.

6.     The specific detectivity, NPDR, and NEP are also important parameters for a photodetector. Authors should measure these parameters and add in the revised manuscript. In addition, the absorption spectra of Ga2O3 and SnOx/Ga2O3 devices should be measured to understand the absorption in the devices.

7.     “----UV-C radiation (C refers to the range 100 ÷ 280 nm of the UV radiation,……” correct the symbol.

8.     English needs to be corrected in the revised manuscript. For instance, “Important is also to detect a target UV-C signal even in presence of daylight, which rules out the employment of standard semiconductors and makes the use of ultra-wide bandgap semiconductors necessary.”

Comments on the Quality of English Language

 English needs to be corrected in the revised manuscript. For instance, “Important is also to detect a target UV-C signal even in presence of daylight, which rules out the employment of standard semiconductors and makes the use of ultra-wide bandgap semiconductors necessary.”

Author Response

RESPONSE LETTER TO THE REVIEWER #1

The manuscript fabricates a Ga2O3-based photoresistor for achieving UV-C by utilizing SnOx at the interface. The authors conducted only electrical measurements. They should also measure the structural and chemical properties, which are very important to design a high-performance photodetector. Overall, the article is not well-organized, and its presentation is also not good. The figure quality and resolution are very poor.

Reply of Authors. Dear Reviewer, thank you for your thorough review and valuable feedback on our manuscript. We appreciate the time and effort you have invested in evaluating our work. However, from your comment, we think that there is a misunderstanding with respect to the goal of our work.

In the present manuscript we did not want to design and fabricate a new photoresistor, with better performance, but rather propose a general approach for the rigorous assessment of the contact role on the overall performance of UV-C photodetectors. We suggest an original method, based on electrical measurements, to assess the interface between the contact and the active material. This is a key parameter that may heavily affect the performance of photoresistors.

The photoresistor studied in this work is based on the k-phase of Ga2O3, a material whose good electrical, structural and chemical properties were well studied and described e.g. in Refs. 16, 32, 38, and 40. Nevertheless, the issue of optimal coupling with the ohmic contacts is still open and the electrical standard methods used to investigate the interfacial energetic configuration are not appropriate and may provide wrong results. The aim of this manuscript is to provide an alternative method that works even in a “difficult context” like that of our case of study.

We improved presentation and organization of the paper, enriching the discussion and adding details and references. The changes in the revised manuscript have been highlighted in red color. The original figures are in high quality PNG format and the scarce quality of them in the PDF file is due to the conversion. Authors will provide high quality figures before the publication. All comments of the Referee were considered as reported below:

  1. The introduction of important concepts in the article must be very rigorous. For the introduction of UV photodetector based on PN junction and MSM, some more references should be added, for instance: (1) Materials & Design 2022, 221, 110917, (2) Advanced Electronic Materials, 2022, 8, 2200392. (3) Journal of Materials Research and Technology 2023, 22, 2174, (4) Solar Energy 2018, 174, 231–239.

Reply of Authors. Although the manuscript is focused on the study of ohmic contacts in MSM photodetectors, we followed the hint of the reviewer to include more references also related to p-n junctions. We have extended the introduction and included some new references as suggested by the Referee, in particular those that focus on the response speed as a function of working device parameters. The sentence below has been added in the Introduction:

“The detection of UV-C light by using gallium oxide was obtained both with use of photodiodes and photoresistors. The time reaction of diodes is normally quicker than that of photoresistors. Furthermore, a photodiode can exploit the photovoltaic effect and operate in self-powered mode, but in case of Ga2O3, where no p-doping is available, p-n junctions require a heterostructure that can adversely affect the carrier collec-tion because of the point defects and traps associated to the interface, interdiffusion phenomena and possible interlayer formation. Schottky diodes also present issues and tend to fail at high temperatures [19-21]. The presence of trapping states at the surface or at the interfaces can reduce drastically the response speed. On the other hand, it should be noted that the intensity of incoming light, the contact geometry, and the ap-plied voltage also play an important role on the photoresponse speed [22,23].

Photoresistors offer some advantages in terms of easier fabrication procedure and higher sensitivity thanks to the larger active volume available for carrier photogeneration. The improvement of contact interface quality is however crucial for the photoresistors, as good low-resistivity ohmic contacts are essential for the effective collection of photocarriers.”

  1. Authors have chosen k-Ga2O3 but it is found that it is not that much stable. The authors need to add more explanation as to why k-Ga2O3 is more important and useful instead of β-Ga2O3.

Reply of Authors. Please note that nowhere in the manuscript it was stated that k-Ga2O3 is more important and useful than β-Ga2O3.  Our group started the research on the k polymorph some years ago and demonstrated that it is thermally stable up to 700 °C, has higher crystallographic symmetry and can be deposited at lower temperature under milder epitaxial conditions. This makes the k-phase of Ga2O3 cost-effective and interesting for non-critical operational conditions, such as for typical UVC detections. Of course, this does not hold for high power electronics in high temperature environments.  

A sentence was added to the Introduction to underline advantages and limits of the k polymorph:

“Our group started the research on the k-polymorph some years ago and demonstrated that it is thermally stable up to 700°C, it has higher crystallographic symmetry than the monoclinic b-Ga2O3, and can be deposited on conventional substrates (like sapphire) at lower temperature and milder epitaxial conditions. A complete conversion of the metastable k-phase to the stable monoclinic β-phase occurs only above 900°C. In addition, it exhibits symmetric electronic and vibrational properties, and no cleavage problems. These characteristics make the k-phase of Ga2O3 cost-effective and interesting for non-critical operational conditions, such as for typical UV-C detection, and justifies the efforts made to overcome the residual shortcomings, via optimization of growth parameters and enhancement of the quality of interfaces [15-18].”

  1. Interface is very important for a photodetector because of charge transport and interfacial defect deteriorates the charge transport. There is a recent research work on the improvement of interface quality of Ga2O3-based diode: Appl. Phys. Lett. 2024, 124, 262101. Authors should compare their diode characteristics with those reported in a Table.

Reply of Authors. We do not follow the reviewer on this question. What should a comparison between the characteristics of a heterojunction diode and a photoresistor add to the philosophy of this work? Even limiting the comparison to the interfaces of the two types of devices makes little sense as they are very different in nature.

  1. Authors should add a digital image of the device having a TLM pattern on it.

Reply of Authors. We agree with the Reviewer that an image of the device can be useful; we added it in the Figure 1.

  1. The rise and fall times of the SnOx/Ga2O3 photoresistor are in the range of seconds, whereas the current photodetectors based on Ga2O3 show ms rise time. The authors add some discussion on it.

Reply of Authors. As already pointed out, this a methodological work that demonstrates that the presence of traps under the contacts of the tested device has deleterious effect on the time response.  In addition, the intensity of incoming light, the contact geometry, and the applied voltage can also affect the photoresponse speed so that a direct comparison with data reported in literature is not straightforward. What is most significant here is the comparison between results obtained on the same Ga2O3 material, by changing distance between contacts and the contact material (see Supporting Information for a comparison between TiAu and SnO2-x contacts).

  1. The specific detectivity, NPDR, and NEP are also important parameters for a photodetector. Authors should measure these parameters and add in the revised manuscript. In addition, the absorption spectra of Ga2O3 and SnOx/Ga2O3 devices should be measured to understand the absorption in the devices.

Reply of Authors. The goal of the present work is not the presentation of an “optimized” photoresistor based on k-Ga2O3. For this reason, we think that is not important to indicate the value of specific detectivity, NPDR, and NEP for our samples. Our intention is now clearly declared in the manuscript:

“The device selected as case study was a photoresistor based on k-Ga2O3 with characteristics sufficiently good to verify the applicability of the new approach as well as its ability to determine the voltage range and contact geometry that give the highest performance. UV-C photoresistors based on this active material demonstrated a rejection ratio UV/VIS higher than 104 and high spectral responsivity as already been reported in [16].”  

The absorption spectra for our films have been already reported in [32]; a sentence has been introduced in the text to give direct information to the reader, also regarding the light transmission on regions of film covered by the contacts:

“Films showed a good optical absorption in the UV-C region with a marked suppression above 270 nm [32], whereas the percentage of UV light transmitted under the contacts is negligible.”

  1. “----UV-C radiation (C refers to the range 100 ÷ 280 nm of the UV radiation ……” correct the symbol.

Reply of Authors. The symbol has been corrected.

  1.    English needs to be corrected in the revised manuscript. For instance, “Important is also to detect a target UV-C signal even in presence of daylight, which rules out the employment of standard semiconductors and makes the use of ultra-wide bandgap semiconductors necessary.”

Reply of Authors. English language was checked in the whole manuscript and amended wherever necessary. Major changes have been highlighted in RED along the text.

The Authors hope to have answered all the Reviewer's questions comprehensively and are confident that after this revision the manuscript meets the requirements for publication.

Best Regards

Maura Pavesi

(Corresponding Author)

Reviewer 2 Report

Comments and Suggestions for Authors

The manuscript is devoted to the study of the influence of contacts on the photoelectric properties of photoresistors based on κ-Ga2O3 films. SnOx was chosen as the contact material for κ-Ga2O3. A methodology for measuring the photoelectric properties of photoresistors based on κ-Ga2O3 films is investigated. The manuscript corresponds to the topic of sensors and is very important for the development of solar blind UVC detectors. The manuscript is well structured and competently written. Before publication, the following minor comments of the reviewer should be taken into account:

1. The abstract should indicate how the photoelectric properties of photoresistors based on κ-Ga2O3 films vary with the type of contacts and their configuration. Give numerical values of the properties.

2. Solar blind UVC detectors based on κ-Ga2O3 are under active investigation. This should be mentioned in the introduction. It is useful to give the characteristics achieved, advantages and disadvantages.

3. Is the deposited film used as contacts polycrystalline? What are its properties, bandgap energy, carrier concentration? Does this film correspond to SnO2-x?

4. Section 3.2 2nd paragraph typo in units of measurement.

5. How was the detector speed performance measured?

6. Figure 8c needs detailed discussion. The reasons for the relationship between the times are unclear.

Author Response

RESPONSE LETTER TO THE REVIEWER #2

The manuscript is devoted to the study of the influence of contacts on the photoelectric properties of photoresistors based on κ-Ga2O3 films. SnOx was chosen as the contact material for κ-Ga2O3. A methodology for measuring the photoelectric properties of photoresistors based on κ-Ga2O3 films is investigated. The manuscript corresponds to the topic of sensors and is very important for the development of solar blind UVC detectors. The manuscript is well structured and competently written.

Reply of Authors. We thank the Reviewer for having appreciated our work and for the time dedicated to the manuscript review. We welcome suggestions that are important for improving the manuscript. A point-by-point response to the Reviewer’s queries is reported below, whereas the changes in the revised manuscript have been highlighted in RED color.

  1. The abstract should indicate how the photoelectric properties of photoresistors based on κ-Ga2O3 films vary with the type of contacts and their configuration. Give numerical values of the properties.

Reply of Authors. The discussion of the influence of type of contact and configuration requires a wider space than that dedicated to the Abstract. A significant comparison among devices with different contacts (type, geometry, configuration) is possible only if we use the same active material for all investigated devices. In our case, the comparison is possible for SnO2-x and TiAu contact (see Supporting Information) because these photoresistors were fabricated with pieces of the same epilayer. The comparison between different contact spacing is particularly significant in our case.

The numerical values of the photoelectrical properties are reported in the main text of the manuscript but not in the Abstract. Note that our goal was not to show that our photodetector has the best performance among Ga2O3-based devices, but rather investigate the interface between the contact and the photosensitive material. It is shown that the electrical characterisation of the interface is tricky and may lead to erroneous interpretation when it is carried out with conventional 2-point I-V measurements.

We added a sentence, particularly regarding the photoresponse speed, in the Introduction:

“On the other hand, it should be noted that the intensity of incoming light, the contact geometry, and the applied voltage also play an important role on the photoresponse speed [22,23]. “

  1. Solar blind UVC detectors based on κ-Ga2O3 are under active investigation. This should be mentioned in the introduction. It is useful to give the characteristics achieved, advantages and disadvantages.

Reply of Authors. Thank you for the suggestion. We have extended the Introduction (in RED color in the manuscript), increasing the emphasis on the current interest in the study of these devices. Further, we added other relevant references.

  1. Is the deposited film used as contacts polycrystalline? What are its properties, bandgap energy, carrier concentration? Does this film correspond to SnO2-x?

Reply of Authors. The contact film was deposited by reactive d.c. magnetron sputtering in an Ar+Oâ‚‚ environment, starting from a Sn target with a purity of 6N. The films deposited at room temperature (RT) are amorphous, and by varying the sputtering power density and the partial pressure of Oâ‚‚, SnOâ‚‚â‚‹â‚“ films were obtained (with 0 ≤ x ≤ 1). For the contacts, slightly Sn-rich films were chosen, characterized by an energy gap of 3.56 eV, a carrier concentration of 1.25×1020 cm−3, and an electron mobility in the range of 1–5 cm²/Vs. Additionally, the mobility increases up to 15 cm²/Vs with increasing oxygen content in the sputtering chamber. These properties correspond to a resistivity in the range of 5×10−2 ¸ 1×10−2 Ω⋅cm.

In the manuscript, this additional information has been added and SnOx has been changed in SnO2-x, as suggested by the Referee, to avoid misunderstanding.

  1. Section 3.2 2nd paragraph typo in units of measurement.

Reply of Authors. The typo has been corrected.

  1. How was the detector speed performance measured?

Reply of Authors. At the end of the section Materials and Method, the description for the acquisition of time transients has been completed by adding the sentence:

A mechanical shutter was used to modulate the light directed to the samples, and get on-off illumination. The off period was shorter than the characteristic photoresponse time of the investigated devices. The time transients after turning the light on and off were acquired with a 66 Hz sampling frequency.”

  1. Figure 8c needs detailed discussion. The reasons for the relationship between the times are unclear.

Reply of Authors. The trend of photoresponse in Figure 8c is discussed in the manuscript and, to support the hypothesis of the influence of the contact interface, a comparison with another type of contact (TiAu) was made in the Supporting Information. The reasons for the relationship between the times are discussed in the manuscript with addition of the text below:

The transient rise time as a function of contact spacing follows a monotonic trend that cannot simply be explained by considering the ohmic equivalent circuit used in the steady state regime. Immediately after illumination the resistance of active layer of Ga2O3 decreases significantly due to carrier photogeneration, which, for a given applied voltage, results in a modified voltage drop at photoactive material and contact regions. The photocurrent transient under illumination depends on the settlement of the space charge under the contacts, up to reaching the final band bending. This evolution must be treated considering that contacts behave as an impedance rather than pure resistance. Therefore, the shorter the contact distance the stronger its effect on the transient because the specific contribution of the two contacts to the total circuit impedance is higher.

The role of the contacts on photoresponse speed is also demonstrated by the different results obtained using the same Ga2O3 absorber but with Ti/Au contacts. Despite the good linearity over a wide bias range (Figure S1 in Supporting Information), the response time to UV illumination for the same pair of contacts is longer than for SnO2-x (Figure S2). This suggests that for some applications the SnO2-x-ITO contact system may be preferable to the Ti/Au one.”

Authors are confident that after this revision the manuscript meets the requirements for publication.

Best Regards

Maura Pavesi

(Corresponding Author)

Round 2

Reviewer 1 Report

Comments and Suggestions for Authors

Authors already measured the photoresistor properties. I didn't understand why this work is not related to fabricating a photoresistor. The authors have studied the effect of metal contact on the electrical properties of a photoresistor, not the effect of the interface, so the title of this manuscript should be changed. There is a difference between studying the effect of interface and the role of contact. They should measure the interfacial defect density to confirm the quality of the electrical contact in the revised manuscript. 

In addition, the authors still didn't provide a satisfactory response to many previous comments. They should provide the structural and chemical properties of materials such as XRD, SEM, and XPS results.

Author Response

Please see attached the file.
